# Polysome Profiling Proves Impaired IL-10 and Caspase-8 Translation in PBMCs of Hemodialysis Patients

**DOI:** 10.3390/biom15030335

**Published:** 2025-02-26

**Authors:** Amanda Dawood, Roman Fiedler, Silke Markau, Matthias Girndt, Christof Ulrich

**Affiliations:** 1Department of Internal Medicine II, Martin Luther University Halle-Wittenberg, 06120 Halle, Germany; amanda.dawood@student.uni-halle.de (A.D.); roman.fiedler@uk-halle.de (R.F.); silke.markau@uk-halle.de (S.M.); matthias.girndt@uk-halle.de (M.G.); 2KfH Nierenzentrum Halle (Saale), 06120 Halle, Germany

**Keywords:** polysome profiling, inflammation, IL-10, caspase-8, hemodialysis, lipid disorder

## Abstract

Triggered by uremic intoxication, a surplus of inflammatory mediators is present in the serum of hemodialysis (HD) patients. Anti-inflammatory counterbalancing mechanisms initiated by interleukin-10 (IL-10) and caspase-8 (Casp-8) appear to be disturbed. Earlier observations let us suppose that translational rather than transcriptional mechanisms are responsible for this effect. Therefore, we investigated the polysome profiling of isolated PBMCs to study gene-specific mRNAs attached to monosomes and polysomes in HD patients (n = 42), patients with lipid disorder and normal renal function (LD, n = 10) and healthy control subjects (CO, n = 9). CRP (C-reactive protein) as a marker of inflammation was significantly elevated in HD and LD patients compared to CO subjects. NGAL (neutrophil-associated lipocalin), a potential marker of kidney disease and inflammation was increased in HD versus LD and CO. LD patients, however, had significantly higher proteosomal IL-10 and Casp-8 activities. LD and HD are two high cardiovascular risk groups with microinflammation. Lower translational activities of IL-10 and Casp-8 mRNAs in HD may be the result of a weak anti-inflammatory response potentially associated with the uremic immune defect.

## 1. Introduction

The balance of local microenvironments in the human body is essential for tissue homeostasis. Inflammation and fibrosis and repair and regeneration mechanisms are shaped by the interplay of inflammatory mediators such as interferon-γ, TNF-α, IL-6 and IL-1 and anti-inflammatory cytokines of the IL-10 family. IL-10 is primarily produced by immune-active cells like monocytes/macrophages, dendritic cells, some T-cell subsets as well as NK- and B-cells. The main function of IL-10 is to control inflammation and to regulate adaptive immune responses [1]. IL-10 signaling is followed primarily by the activation of the transducer and activator of the transcription family (STAT), leading to anti-inflammatory, immune-modulatory and anti-apoptotic effects [2]. However, there are also reports demonstrating that high serum levels of IL-10 induce T-cell apoptosis [3].

It is of note that a process initiated by caspase-8 (Casp-8) is also considered anti-inflammatory because Casp-8 activation indirectly leads to the elimination of defective and/or potentially dangerous cells by local phagocytes [4]. Anti-inflammatory mechanisms are the essential means to counterbalance inflammatory reactions, which are often part of chronic diseases. A permanent low-grade inflammation because of uremic intoxication is characteristic of hemodialysis (HD) patients. Uremia is defined by the accumulation of inflammatory cytokines and toxic, nitrogenous compounds that are normally excreted by intact kidneys. Consequently, high CRP—a marker of inflammation—and low IL-10 levels are found in patients with end-stage renal failure [5]. The clinical sequelae followed by inflammation and immune dysfunction have a high prevalence in cardiovascular complications (e.g., atherosclerosis) and infections [6,7]. For a long time, it has been known that atherosclerosis is an inflammatory disease that intertwines lipid metabolism, inflammation and endothelial damage [8]. Thus, there remain questions about the significance of uremia/cardiovascular disease regarding Il-10 production. Do patients with a cardiovascular risk profile but intact kidneys have a low IL-10 status, too? To approach this question, we included patients with lipid disorder (LD) but healthy kidneys in our pilot study. Furthermore, there remains to be proven why dialysis patients with a high inflammatory burden cannot adequately counterbalance the high level of inflammatory IL-6. We already addressed this question at least in part and found discrepancies between measured IL-10 transcription rates and translated proteins [9]. Polysome profiling can give a distinct indication about the translational status of a specific mRNA [10]. Polysome profiling is based on the sucrose gradient separation of translated mRNAs, which are associated with monosomes (80S ribosomes) or polysomes (two to several ribosomes per mRNA). Therefore, translation is governed by the dynamic distribution of mRNAs into non-translating ones since the former are either free or associated solely with the 40 S subunit while the latter are associated with one to several ribosomes [11]. Given that actively translated mRNAs are often associated with multiple ribosomes, the ratio of polysome to monosome fractions can be used as a measure of the translational status and to evaluate the efficiency of this process [12].

Regarding the inflammatory nature of cardiovascular-related diseases, it is essential to integrate CRP measurement in the study. NGAL appears to be another valuable tool in characterizing our patient groups. NGAL was originally studied as a biomarker in acute kidney failure as it is released after tubular injury [13]. However, NGAL is mainly produced by macrophages and neutrophils after an inflammatory insult. Thus, NGAL is increased in chronic kidney disease, too [14]. Moreover, NGAL is also related to lipid disorders and cardiovascular-related diseases [15].

Therefore, in this study, we will shed light on the translational profile of the anti-inflammatory genes IL-10 and Casp-8 in two high-cardiovascular-risk groups with an inflammatory background, i.e., patients with lipid disorders and patients on maintenance hemodialysis in comparison to healthy controls.

## 2. Materials and Methods

### 2.1. Study Population

This cross-sectional pilot study included 3 cohorts. Patients with lipid disorder (LD, n = 10) were routinely checked at the Department of Internal Medicine II, healthy controls (CO, n = 9) were recruited from the medical staff of the university hospital of Halle, and hemodialysis patients (HD, n = 42) were recruited from the nephrology outpatient dialysis center of the Department of Internal Medicine II of the University Halle-Wittenberg. Inclusion criteria of HD patients included age > 18 years and a history of hemodialysis treatment > 12 weeks. LD and CO subjects had to be >18 years of age. For all study subjects, patients with active malignancy, active infections and neurological disorders were excluded from the study. All patients gave their informed consent. Patients with lipid disorder had severe hypercholesterolemia with (n = 2) or without (n = 8) manifest cardiovascular disease. None of them had impaired kidney function (estimated glomerular filtration rate (eGFR) < 60 mL/min. Reasons for kidney failure included vascular nephropathy (23.8%), diabetic nephropathy (14.8%), glomerulonephritis (19.0%), interstitial nephritis (7.1%), ADPKD (14.8%) and others (21.4%). All immunobiological samples of HD patients were taken after a long intradialytic interval and before the start of the first dialysis session of the week. Blood samples of LD and CO were collected in the morning. The study was conducted according to the Declaration of Helsinki. Written informed consent was obtained from all study subjects, and the study protocol was approved by the local ethics committee.

### 2.2. Absolute Cell Count Determination

Trucount tubes (BD Biosciences, Heidelberg, Germany) were used to determine absolute cell count numbers in 50 µL of whole blood. Total leucocytes were identified by anti-CD45 (Miltenyi Biotec, Bergisch-Gladbach, Germany), granulocytes by -CD15 (Thermo Fisher Scientific, Darmstadt, Germany), monocytes by -CD14 (Milentyi Biotec) and lymphocytes by -CD3 (Miltenyi Biotec), -CD4 (Biolegend, Koblenz, Germany), -CD8 staining (Miltenyi Biotec).

### 2.3. PBMC Isolation

PBMCs were isolated by Ficoll density centrifugation (GE Healthcare, Solingen, Germany) from EDTA blood samples drawn from the dialysis access before the dialysis session. The quality of isolated cells was tested by 7-AAD staining. The vitality of PBMCs was 99.3% ± 0.7 for CO, 99.7% ± 0.3 for LD and 99.5% ± 0.5.

### 2.4. Stimulation of PBMCs

In total, 0.8–1.0 × 10^7^ cells were each incubated in 25 cm^3^ tissue culture flasks (TPP, Trasadingen, Switzerland). One aliquot was stimulated with 10 µg/mL of lipopolysaccharide (LPS (0127: B8, Sigma-Aldrich, Steinheim, Germany)). After 16 h under 5%CO_2_ at 37 °C, the flasks were treated with cycloheximide (CHX; 100 µg/mL, Sigma-Aldrich) for 15 min.

### 2.5. Preparation of Cytosolic Lysates

After scrapping out the tissue flasks, the cells were washed using ice-cold PBS. Afterwards, the cells were lysed on ice for 30 min using a cycloheximide containing lysis buffer (5 mM Tris-Base, 1.25 mM MgCl_2_, 1.5 mM KCl, 0.5% sodium-deoxycholate (Sigma-Aldrich), 2 mM DTT (Roth, Karlsruhe, Germany), 0.5% Triton-X-100 (Roth), RNase-out (Thermo-Fisher Scientific) and 100 µg/mL CHX). Prior to use, the lysis buffer was supplemented with 6 µL/mL RNaseOut (Thermo Fisher Scientific). An aliquot of the lysed suspension was taken for RNA extraction (the mRNA sample before gradient preparation).

### 2.6. Preparation of Linear Sucrose Gradients

In total, 5 and 45% sucrose solutions were prepared in polysome buffer (5 mM Tris-Base, 1.25 mM MgCl_2_, 1.5 mM KCl, RNase-out and 100 µg/mL CHX); a layering device (BioComp, ScienceServices GmbH, München, Germany) was used to fill the open-top polyclear ultracentrifuge tubes (13 × 51 mm, Seton Scientific, München, Germany) with the sucrose solutions, followed by linear gradient formation using the gradient maker (BioComp). Afterwards, the samples were applied to the corresponding tubes. The tubes were sealed with rate zonal caps (BioComp), placed in a precooled MLS 50 rotor (Beckman Coulter, Krefeld, Germany) and centrifuged at 130,000× *g* for 1.5 h. Afterwards, 430 µL fractions were collected with the Piston Gradient Fractionator (BioComp). The process was UV-monitored (integrated UA6 UV monitor) using BioComp software version 2.10.00. Fraction 5 was identified as “the monosomal fraction”; followed by fraction 6, which was determined as the “early polysomal fraction”; and fraction 9 was defined as the “late polysomal fraction”.

### 2.7. RNA/cDNA/qPCR

RNA was isolated from PBMC lysates using the Direct-zol MiniPrep Plus Isolation Kit (ZymoResearch, Freiburg, Germany). The RNA concentration and quality of the different fractions were tested by the Z-Tecan Infine^®^200 Pro technique (Tecan Group, Männedorf, Germany) (given in Table 1).

Equal amounts of RNA (45 ng) were reverse transcribed using the High-Capacity cDNA Reverse Transcription Kit (Thermo Fisher Scientific, Darmstadt, Germany).

IL-10 (Hs00961622_m1), Casp-8 (Hs01018151_m1), and RPL37A (Hs01102345_m1) mRNA expression were analyzed using TaqMan probes (Thermo Fisher Scientific) and qPCRBIO Probe Mix High-ROX (Nippon Genetics, Düren, Germany). The samples were processed in duplicates on a StepOnePlus cycler (Thermo Fisher Scientific). The data were normalized by RPL37A (the dCt method).

### 2.8. Cytokine Analysis

CRP and NGAL were analyzed in the serum using ELISA techniques (CRP, Biomol, Hamburg, Germany and NGAL, Fisher Scientific, Schwerte, Germany). The data were analyzed on the ELX808 microplate reader (Bio-Tek Inc., Berlin, Germany).

### 2.9. Statistics

The results are expressed as mean ± SD. All continuous variables were controlled for normal distribution using the D’Agostino–Pearson omnibus test. Continuous data were compared by one-way ANOVA followed by Friedman or Tukey’s post test as appropriate. Categorical variables were analyzed by the chi-square test. All calculations were carried out using SPSS 21.0 (SPSS Inc., Chicago, IL, USA) or GraphPad Prism 9.2.0 statistics software (GraphPad Software Inc., La Jolla, CA, USA). The level of significance was set at *p* < 0.05.

## 3. Results

### 3.1. Demographic Data

To classify the anti-inflammatory capacity of HD patients, we included two control groups. On the one hand, a relatively young population of healthy subjects (CO) was included in the study, and on the other hand, we chose patients with lipid disorders (LDs) but intact kidney function. This group shares some atherosclerotic risk profiles with HD patients. Most prominently, arterial hypertension characterizes LD and HD patients (Table 2). In contrast to healthy controls, LD and HD patients were not different in age, gender and the prevalence of diabetes (Table 2). Healthy control subjects had a significantly lower level of the inflammatory marker CRP in comparison to LD and HD. HD patients tended to have the highest CRP levels in the three groups. NGAL, a marker of residual renal function and inflammation, was significantly increased in the serum of HD patients in comparison to CO and LD patients.

### 3.2. Cellular Parameters

Blood cell analysis revealed that the three groups did not differ regarding granulocyte and monocyte numbers. However, lower lymphocyte numbers, in particular CD4+ numbers, were detected in HD patients (Table 3).

### 3.3. Translational Efficiency Measured by Polysome Profiling

Polysome profiling is a method to fractionate different RNA/ribosome fractions, including monosomes and early and late polysomes (Figure 1).

The determination of the area under the curve (AUC) gives the first impression of the translational activity of CO, LD and HD.

The “monosomal activity” of LD and HD patients was significantly lower under basal and stimulated conditions compared to CO subjects (Figure 2a,b). Similar findings were observed for the total “polysomal activity” under basal conditions (Figure 2c). Upon LPS stimulation, a slight shift to higher AUCs was observed in LD patients (Figure 2d). The ratio of polysomal to monosomal AUCs gives a good estimate of the “net translational efficiencies” in the three groups. As demonstrated in Figure 2e,f, the significantly highest translational activity is observed in LD patients, while the AUCs of HD patients are at a similar level to those of the CO subjects (Figure 2e,f).

### 3.4. Course of IL-10 Expression in the Different Monosomal/Polysomal Fractions

Analyzing the monosomal and polysomal IL-10 translation in the three study cohorts, we find a trend in the enrichment of the different fractions compared to the individual control sample taken before gradient centrifugation (bG, before gradient preparation). This observation especially applies to HD patients (Figure 3c) where we find enriched IL-10 translational activity throughout the polysomal fractions. Because of the low number of study participants, this effect is not as strongly developed for CO and LD (Figure 3a,c). But in healthy controls, there seems to be an enrichment of IL-10 mRNA in the monosomal (F5), early polysomal (F6) and late polysomal fraction (F9) (Figure 3a), while the pattern of enrichment in LD patients resembles the enrichment of IL-10 mRNA in HD (Figure 3b,c). The LPS challenge of PBMCs enhanced IL-10 mRNA expression in all study subjects. In CO subjects, no significant IL-10 enrichment in the different fractions was observed (Figure 3d). In LD patients, the highest enrichment level was seen in fraction 8 (Figure 3e), while in HD patients, the late polysomal fraction F9 showed the highest IL-10 mRNA translational activity (Figure 3f).

### 3.5. Course of Casp-8 Expression in the Different Monosomal/Polysomal Fractions

Although the translational Casp-8 pattern is quite different between the three cohorts (Figure 4a–c), the highest Casp-8 translation is mostly measured in the monosomal (F5) or early polysomal fraction (F6). This is notably the case for CO (Figure 4a) and HD (Figure 4c) patients, whereas high translational Casp-8 activity is measured in all fractions of LD patients (Figure 4b). Interestingly, the LPS challenge of PBMCs had only a minor influence on Casp-8 mRNA translation. Thus, basal Casp-8 mRNA levels and expression after LPS stimulation did lead to similar Casp-8 mRNA enrichment in the respective fractions (CO, Figure 4a,d; LD, Figure 4b,e; HD, Figure 4c,f).

### 3.6. Comparison of Monosomal and Polysomal IL-10 Translational Activity in Different Cohorts

In this analysis, we focused on the monosomal fractions and early and late polysomal fractions. This investigation was chosen to discriminate between translational products that were translated by one, two or up to five ribosomes. As can be seen in Figure 5a, initially (before gradient preparation), there was no significant difference in IL-10 mRNA expression between CO, LD and HD. However, there was a slight trend in higher IL-10 translational activity in LD patients. This trend is obvious in the monosomal fraction (Figure 5b) too. In the early polysomal fraction, LD patients have a significantly higher translational IL-10 activity, while the mRNA levels of CO and HD patients remain on the same level (Figure 5c). IL-10 translational activity in the late polysomal fraction (Figure 5d) reflects the expression situation as seen in the monosomal fraction. Although the IL-10 translation was greatly increased by LPS stimulation (Figure 5e–h), it seems that this kind of stimulus similarly provoked IL-10 production in the three patient cohorts.

### 3.7. Comparison of Monosomal and Polysomal Casp-8 Translational Activity in Different Cohorts

Figure 6a demonstrates that LD patients have higher Casp-8 translational activity when analyzed before gradient preparation. It is of note that compared to the bG-fraction, both the monosomal (Figure 6b) and the early polysomal fraction (Figure 6c) show a significant drop in Casp-8 dCt values, independent of which study group is analyzed and whether the sample is stimulated (Figure 6f,g) or not. Noteworthy, healthy controls (CO) showed a significantly higher expression of Casp-8 compared to HD patients. The most interesting observation, however, is that LD patients have significantly higher Casp-8 expression versus CO and HD patients in the “late polysomal” fraction. This holds true for the unstimulated (Figure 6d) and stimulated samples (Figure 6h).

### 3.8. Influence of Dialysis-Related Parameters with Regard to Monosomal/Polysomal IL-10 and Casp-8 Expression

Dialysis-related parameters like dialysis vintage, Kt/V, and the kind of dialysis access (arteriovenous fistula versus catheter) may have an impact on monosomal/polysomal IL-10/Casp-8 expression. Upon analyzing the data this way, we did not see any effect. Neither dialysis vintage, Kt/V nor the kind of dialysis access were related to IL-10 or Casp-8 expression (see Appendix A).

## 4. Discussion

HD patients are in a pro-inflammatory state, resulting in cardiovascular disease [16,17]. Anti-inflammatory counterbalance seems to be insufficient in these patients [18]. Patients with lipid disorders are also confronted with inflammation and are at a high risk for cardiovascular diseases [19,20,21]. Our data confirm that microinflammation is present in HD and LD patients, whereby a trend of higher CRP values is found in HD patients. NGAL (Lipocalin-2) is discussed in the literature in different ways [22]. Principally, high NGAL seems to be associated with acute kidney failure, but considering chronic kidney disease, NGAL levels rise due to a decrease in glomerular filtration rates. It is suggested that inflammation per se triggers lipocalin-2 expression, and there are also reports that describe the association between NGAL serum concentrations and lipid disorders [23]. But in our small cohort, we do not find elevated NGAL levels in LD patients in comparison to healthy controls. In contrast, HD patients have 8 to 10 higher NGAL serum levels in comparison to CO and LD. Thus, we do not believe that NGAL primarily reflects the inflammatory state but rather is a function of tubular damage. Therefore, our NGAL analysis confirms that both CO and LD subjects are not affected by functional kidney problems.

Given that inflammation is present in HD and LD patients, there is interest in how far anti-inflammatory molecules are regulated in both groups. IL-10 is mainly produced by immune cells and a dysregulation of IL-10 is proposed for HD patients. In particular, in HD patients, reduced IL-10 protein production was observed while detecting normal IL-10 transcripts by rt-PCR [24]. Regarding the anti-inflammatory repertoire, one should clearly keep in mind that not only IL-10 but also other mediators may exert anti-inflammatory effects. Often, such mediators are “Janus-like”, which means that depending on the environmental setting, they play deleterious (inflammatory) or protective (anti-inflammatory) roles. IL-6, for example, is significantly elevated in HD patients [25], and it is well accepted that, on the one hand, IL-6 can enhance inflammation, but on the other hand, i.e., in the absence of the suppressor of cytokine signaling 3 (SOCS3), it can induce an anti-inflammatory response [26].). In this paper, however, we focused on the pure prototypical, anti-inflammatory cytokine IL-10.

Polysome profiling is a powerful tool for the analysis of mRNA translation—this technique distinguishes between highly translated mRNAs and those with low translational status [10]. Thus, this kind of analysis avoids the disadvantage of common mRNA analysis that reports gene-specific mRNAs regardless of the translational activity. It is known that the mRNA steady state is only weakly associated with the composition of the proteome [27].

The area under the curve analysis (AUC) of polysome profiling experiments may give a general overview about the translation efficiency of the three cohorts in our study. It is of note that the monosomal and polysomal AUCs are highest in healthy controls, while the AUCs of HD and LD do not significantly differ. However, the ratio of polysomal to monosomal AUCs is a good way to describe translational activity [12]. Analyzing our data, we see significantly higher translational activity in LD patients compared to the two other groups. More astonishingly HD and young and healthy controls do not have different AUC ratios. This is of importance because ribosomal levels actively regulating translation would be expected to have higher general translational activity compared to young control subjects. But perhaps the so-called uremic immune defect limits several translational activities [28,29]. For a long time, it has been known that both monocytes and T- and B cells are impaired by uremic intoxication [30]. A reduction in T-cell numbers, also found in this study, is characteristic for HD [31].

Regarding IL-10 translation, we see that the “early polysomal” fraction of LD patients has higher “IL-10 translational activity”. Unfortunately, we did not save PBMC culture supernatants to measure IL-10 protein levels to check the IL-10 content in PBMCs by a different method. But in general, polysome profiling in combination with gene-specific qPCR is a reliable method that gives a good estimate about the level of protein being expected. Interestingly, stimulation with a low dose of LPS increased IL-10 expression in all samples to the same extent. This suggests that, firstly, the LPS response is functional in all cohorts, and secondly, because of a generally higher IL-10 expression in LD patients, other molecules than LPS must drive IL-10 expression in LD patients. However, we also must be aware that components of Gram-negative bacteria (LPS) probably are not the best way to provoke an IL-10 response because most HD patients struggle with Gram-positive bacteria (Staphylococci) in the first line [32,33].

The second gene of interest in our study was Casp-8. This cysteine-aspartate-specific protease has an important role in cell homeostasis and has key functions in the immune system [34]. Furthermore, it is central in the crosstalk of cell death and inflammation [35]. In patients with kidney disease, apoptosis and necrosis are important in the pathogenesis of many different kidney diseases [36], but undoubtedly, Casp-8 is also involved in cellular processes concerning CVD patients with lipid disorders [37].

As observed for IL-10 translation, LD patients show higher Casp-8 translational efficiency compared to HD and CO. This was most distinctive in the late polysomal fraction. LPS stimulation seems to have no effect on Casp-8 translation. Stimulated and unstimulated samples did not differ.

It is known that cells balance monosome and polysome levels during cellular stress, increasing the activity of polysomes [38]. However, we find only higher Casp-8 expression in LD but not in HD compared to healthy subjects. Casp-8 is a double-edged sword as, on the one hand, it initiates the removal of unwanted cells, and on the other hand, it can, in the case of overshooting responses, lead to the detrimental loss of essential cells [36]. The observation that LD patients have higher Casp-8 expression compared to HD cannot explicitly be explained. It is possible that lower Casp-8 translation activity in HD is part of the uremic immune defect scenario, but possibly, the observed effect also represents a protective mechanism that prevents overshooting reactions.

## 5. Conclusions

Our study sheds new light on a dysregulated anti-inflammatory response in HD patients. Our data reveal a very “naïve” anti-inflammatory translational profile in HD patients resembling the anti-inflammatory response of young and healthy subjects. Thus, together with dysregulated STAT3 signaling and the putative impact of miR-142-3p on IL-10 regulation, low translational IL-10 activity may prevent the effective resolution of inflammation and removal of damaged cells.

Of course, we are aware that our study is small. Greater patient cohorts are necessary to confirm these preliminary results. Another drawback of this study is the lack of IL-10 protein analysis and Casp-8 activity measurement in the cell culture supernatants to confirm the data found in polysome profiling. But nevertheless, polysome profiling is a very important technique which will help us to better understand translational regulatory mechanisms in HD patients.

## Figures and Tables

**Figure 1 biomolecules-15-00335-f001:**
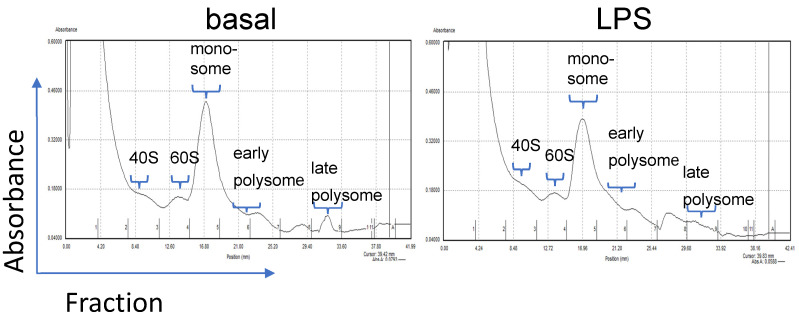
Representative figures of polysome separation curves in unstimulated and stimulated (LPS) PBMC samples of an HD patient.

**Figure 2 biomolecules-15-00335-f002:**
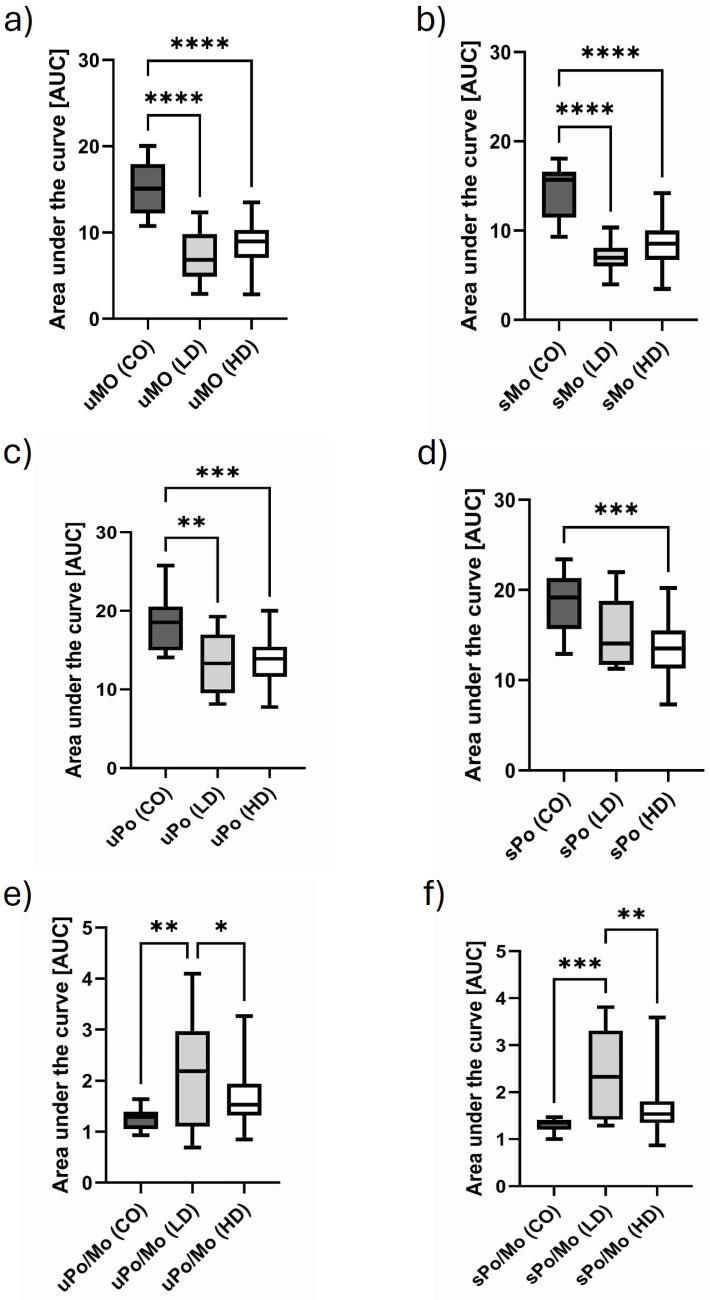
Monosomal and total polysomal activity as determined by the area under the curve analysis. (**a**) Monosomal translational activity (Mo) under unstimulated (u) and (**b**) stimulated (s) conditions. Basal polysomal activity (Po) is shown in (**c**), and the stimulated Po activity is depicted in (**d**). The ratio of polysomal to monosomal translational activity (Po/Mo) as a means of the “total net translational activity” under unstimulated and stimulated conditions is shown in (**e**,**f**). The data are presented as box plots with the median and the 25/75 percentile. The data are analyzed using one-way ANOVA with Tukey’s multiple comparison test as the post test. * *p* < 0.05, ** *p* < 0.01, *** *p* < 0.001, **** *p* < 0.0001.

**Figure 3 biomolecules-15-00335-f003:**
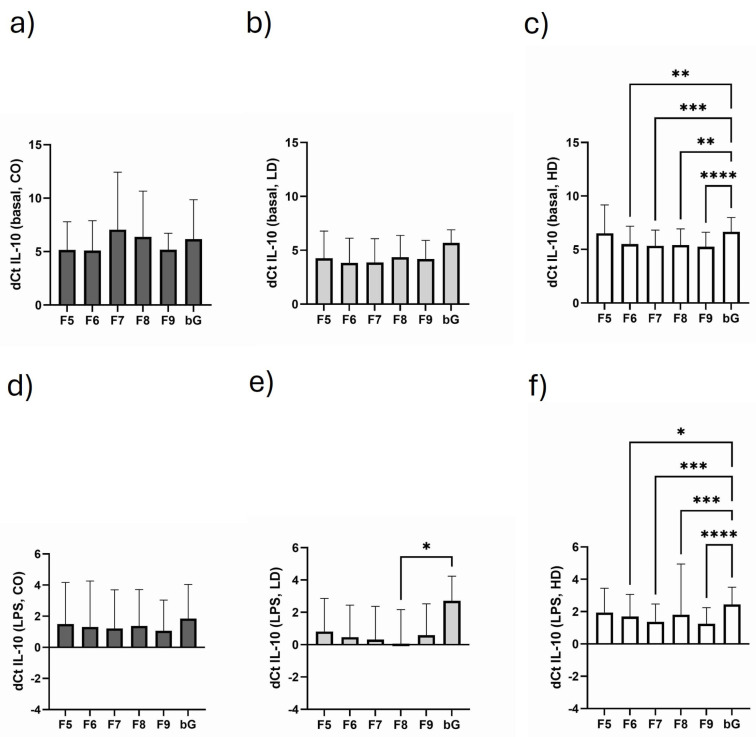
Fractional course of IL-10 mRNA expression under basal (**a**–**c**) and LPS-stimulated conditions (**d**–**f**) in CO (**a**,**d**), LD (**b**,**e**) and HD (**c**,**f**). Data are presented as mean ± SD. Data are analyzed using one-way ANOVA followed by Friedman post test. * *p* < 0.05, ** *p* < 0.01, *** *p* < 0.001, **** *p* < 0.0001.

**Figure 4 biomolecules-15-00335-f004:**
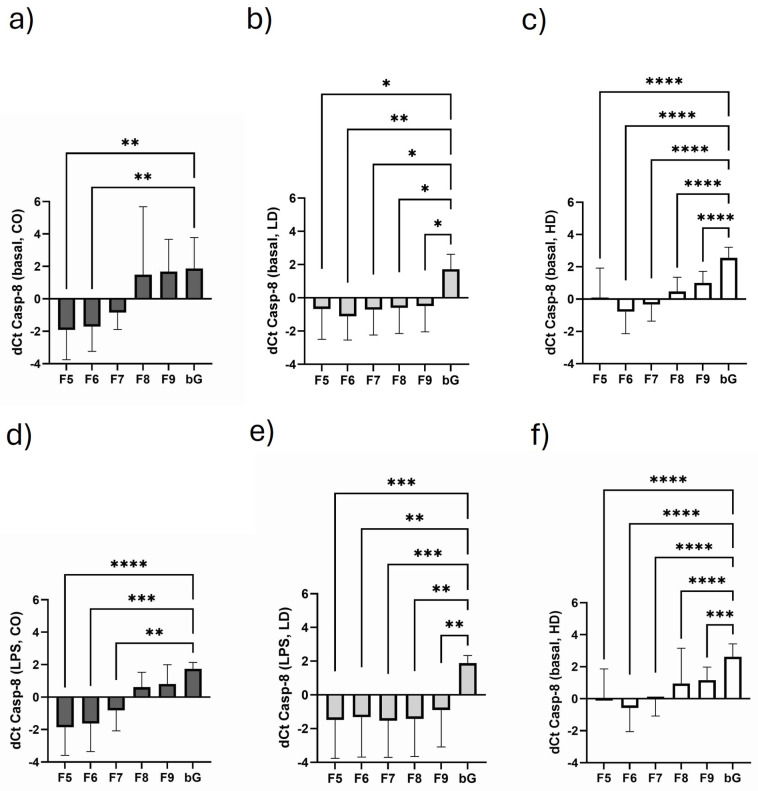
Fractional course of Casp-8 mRNA expression under basal (**a**–**c**) and LPS-stimulated conditions (**d**–**f**) in CO (**a**,**d**), LD (**b**,**e**) and HD (**c**,**f**). Data are presented as mean ± SD. Data are analyzed using one-way ANOVA followed by Friedman post test. * *p* < 0.05, ** *p* < 0.01, *** *p* < 0.001, **** *p* < 0.0001.

**Figure 5 biomolecules-15-00335-f005:**
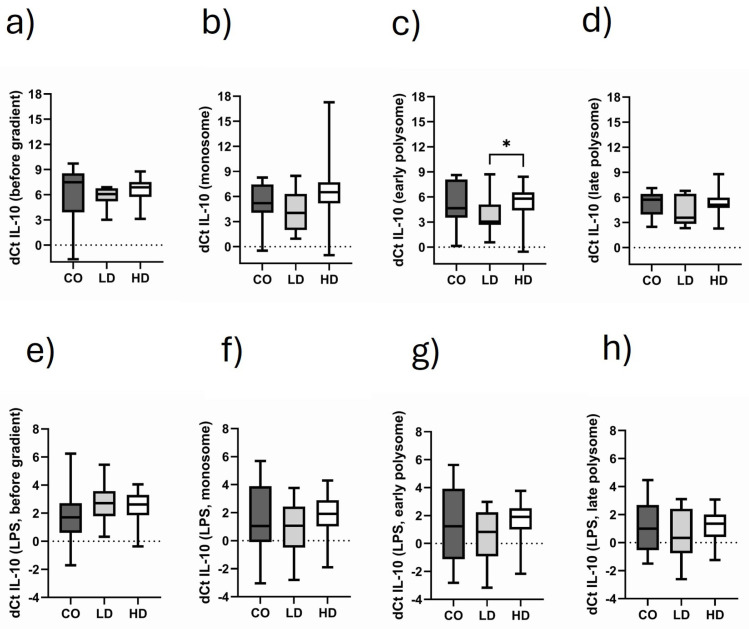
Fractional IL-10 translational activity in CO, LD and HD patients. dCt IL-10 values as analyzed under basal conditions before gradient preparation (**a**) in the monosomal fraction (**b**), the early polysomal fraction (**c**) and the late polysomal fraction (**d**). (**e**–**h**) The corresponding data under stimulatory conditions. Data are presented as box plots with the median and the 25/75 percentile. Data are analyzed using one-way ANOVA with Tukey’s multiple comparison test as posttest. * *p* < 0.05.

**Figure 6 biomolecules-15-00335-f006:**
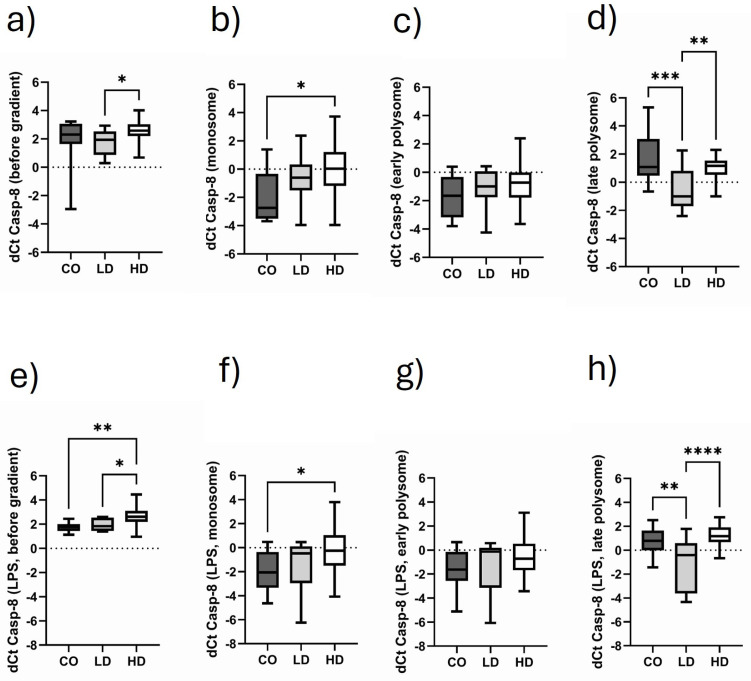
Fractional Casp-8 translational activity in CO, LD and HD patients. dCt Casp-8 values as analyzed under basal conditions before gradient preparation (**a**) in the monosomal fraction (**b**), the early polysomal fraction (**c**) and the late polysomal fraction (**d**). (**e**–**h**) The corresponding data under stimulatory conditions. Data are presented as box plots with the median and the 25/75 percentile. Statistics are performed using one-way ANOVA with Tukey’s multiple comparison test as the post test. * *p* < 0.05, ** *p* < 0.01, *** *p* < 0.001, **** *p* < 0.0001.

**Table 1 biomolecules-15-00335-t001:** RNA quality (260/280 ratio) of the different isolated fractions.

Basal	bG	F5	F6	F7	F8	F9
CO	1.7 ± 0.2	1.9 ± 0.1	1.8 ± 0.1	1.7 ± 0.1	1.7 ± 0.2	1.6 ± 0.2
LD	1.8 ± 0.6	1.9 ± 0.1	1.8 ± 0.1	1.7 ± 0.2	1.7 ± 0.2	1.6 ± 0.2
HD	1.7 ± 0.1	1.9 ± 0.1	1.8 ± 0.2	1.7 ± 0.2	1.7 ± 0.1	1.6 ± 0.3
Stimulated						
CO	1.6 ± 0.2	1.9 ± 0.1	1.8 ± 0.1	1.7 ± 0.1	1.6 ± 0.1	1.7 ± 0.1
LD	1.6 ± 0.3	1.9 ± 0.1	1.8 ± 0.2	1.7 ± 0.2	1.8 ± 0.5	1.7 ± 0.4
HD	1.7 ± 0.3	1.8 ± 0.1	1.7 ± 0.2	1.6 ± 0.2	1.6 ± 0.2	1.6 ± 0.3

Data are presented as mean ± SD. bG: before gradient preparation.

**Table 2 biomolecules-15-00335-t002:** Basic characteristics of the study groups.

	CO (n = 9)	LD (n = 10)	HD (n = 42)	Statistics
Age (years)	37.0 ± 16.8	55.0 ± 7.8	59.6 ± 15.0	* CO vs. LD *** CO vs. HD
Gender (female, %)	33.3	50.0	41.5	n.s.
Diabetes (%)	0	30.0	28.6	n.s.
Sys. BP (mm Hg)	-	147 ± 23	149 ± 18	n.s.
Diast. BP (mm Hg)	-	89 ± 9	79 ± 12	* LD vs. HD
CRP (mg/L)	0.9 ± 0.6	12.5 ± 6.7	23.1 ± 15.3	**** CO vs. HD * CO vs. LD
NGAL (ng/L)	219.0 ± 76.5	270.0 ± 145.4	2333.0 ± 742.6	**** CO vs. HD **** LD vs. HD

Data are presented as mean ± SD. * *p* < 0.05, *** *p* < 0.001, **** *p* < 0.0001. n.s.: not significant.

**Table 3 biomolecules-15-00335-t003:** Cell numbers of immune cells.

	CO (n = 9)	LD (n = 10)	HD (n = 42)	Statistics
Leukocytes (×10^6^/mL)	9.3 ± 2.4	9.8 ± 2.9	9.6 ± 3.0	n.s.
Granulocytes (×10^6^/mL)	4.2 ± 0.4	5.7 ± 2.3	5.6 ± 2.2	n.s.
Monocytes (×10^6^/mL)	0.6 ± 0.1	0.74 ± 0.08	0.71 ± 0.04	n.s.
Lymphocytes (×10^6^/mL)	1.6 ± 0.4	1.7 ± 0.7	1.3 ± 0.6	* LD vs. HD
CD4+ (×10^6^/mL)	0.9 ± 0.4	1.1 ± 0.4	0.7 ± 0.3	* LD vs. HD
CD8+ (×10^6^/mL)	0.5 ± 0.3	0.5 ± 0.4	0.3 ± 0.3	n.s.

Data are presented as mean ± SD. * *p* < 0.05. n.s.: not significant.

## Data Availability

The original contributions presented in this study are included in the article/Appendix A. Further inquiries can be directed to the corresponding author(s).

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
