# Peer review of "Polysome Profiling Proves Impaired IL-10 and Caspase-8 Translation in PBMCs of Hemodialysis Patients"

_biomolecules, 2025, doi:10.3390/biom15030335_

Round 1
Reviewer 1 Report
Comments and Suggestions for Authors
The manuscript entitled “Polyamine profiling proves impaired IL-10 and caspase-8 translation in PBMCs of hemodialysis patients” clearly shows the lower translational activation of IL-10 and caspase-8 mRNAs in hemodialysis patients (HD). The results are very informative.
However, the reviewer recommends slight modifications, if possible.
1. As authors mention in Discussion that the numbers of LD patients and control subjects (CO) are few. If possible, please increase the numbers of LD and CO. In addition, the average age of CO is too young.
2. Please mention in more detail that IL-10 is better than IL-6.
3. Line 201: LD patients → LD patients (Fig. 2d).
4. Line 252: figure 5a → Fig. 5a
Author Response
Response to Reviewer 1
Point 1: The Reviewer states that there are only few control and LD patients included in the study - if possible, their numbers should be increased. Further on, he remarks that the control subjects are much younger compared to the other study groups
Response 1: The Reviewer is right in his statement. However, this study was planned as a pilot project (see Material and Method section, lines 78 - 82) to gain insight of the putative translational dysregulation evident in haemodialysis patients (HD) patients. So, the cardiovascular risk group (LD) and healthy controls (CO) were integrated in the study to get some ideas in how far such a dysregulation is a realistic point to study. Thus, future studies are necessary to deepen the knowledge going along with our new results. About the low average age of control patients, we do not think that this is a blemish of our study for the conclusion derived from the results appears quite clear. The anti-inflammatory repertoire of HD in comparison to the young controls is not as different as would have been expected. Although inflammation is present in HD compared to CO at a much higher extent immune responsive cells of HD are not as potent to adequately counterbalance the inflammation. Therefore, we stated that the cellular anti-inflammatory behaviour is somewhat “naïve” resembling the immune response of even young and healthy subjects (see Conclusion, lines377 - 378).
Point 2: The Reviewer suggests discussing why IL-10 is “better” than IL-6 or why IL-10 has a higher anti-inflammatory impact compared to IL-6, respectively.
Response 2: We agree to the Reviewer that IL-6 can also exert some anti-inflammatory effects (Magno et al, 2019, Yasakawa et al., 2003) but in contrast to IL-10 IL-6 is a profound inflammatory trigger. We added the following sentences to the discussion. Lines 316 - 324.
“Regarding the anti-inflammatory repertoire one should clearly keep in mind that not only IL-10, but also other mediators may exert anti-inflammatory effects. Often such mediators are “Janus- like”, that means depending on the environmental setting they play deleterious (inflammatory) or protective (anti-inflammatory) roles. IL-6 for example is significantly elevated in HD patients (Magno et al.) and it is well accepted that on the one hand IL-6 can enhance inflammation but on the other hand, i.e. in the absence of the suppressor of cytokine signalling 3 (SOCS3), it can induce an anti-inflammatory response (Yasukawa et al.). In this paper, however, we focused on the pure prototypical, anti-inflammatory cytokine IL-10.”
The following references were added to the reference list:
25) Magno, A.L.; Herat, L.Y.; Carnagarin, R.; Schlaich, M.P.; Matthews, V.B. Current Knowledge of IL-6 Cytokine Family Members in Acute and Chronic Kidney Disease. Biomedicines 2019, 7, doi:10.3390/biomedicines7010019.
26) Yasukawa, H.; Ohishi, M.; Mori, H.; Murakami, M.; Chinen, T.; Aki, D.; Hanada, T.; Takeda, K.; Akira, S.; Hoshijima, M.; et al. IL-6 induces an anti-inflammatory response in the absence of SOCS3 in macrophages. Nat. Immunol. 2003, 4, 551–556, doi:10.1038/ni938.
Point 3: The Reviewer remarks that the label “Fig. 2d is missing
Response 3: We are sorry for this mistake and added the corresponding label. Line 201.
Point 4: The Reviewer indicates that figure 5a should be replaced by Fig. 5a
Response 4: We thank the Reviewer pointing to this and replaced the term. Line 252.

Reviewer 2 Report
Comments and Suggestions for Authors
Dear authors thank you very much for allowing me to review your work. You describe that HD patients have lower monosomal and polysomal proportions in their serum. This is accompanied with a lower Il10 transcription level as well as Casp8. The work is novel.
I have the following comments:
- Longer Dialysis vintage is associated with adverse cardiovascular disease outcomes. Is it possible to stratify HD patients (dichotomous stratification) according to dialysis vintage & casp8 and il10 transcription level?
- Could you stratisfy HD patients according kt/V? Dialysis access (AvF/catheter)? Method of Dialysis (online, HD)? Is there any difference for the parameters under study?
- Could you compare samples from peritoneal Dialysis for the parameters tested?
- Is there the possibility to stratisfy the patients under study according to their ejection fraction? If yes, is there a correlation for the paramaters under study?
Minor : Page 5 : The image quality should be better. X an Y axes values are not readable.
All the best
Author Response
Response to Reviewer 2
Point 1: The Reviewer asks for a dichotomous stratification according to dialysis vintage and IL-10/casp-8 expression level.
Response 1: This is an interesting point because as stated by the Reviewer the dialysis vintage may be associated with adverse cardiovascular disease outcomes. We thus stratified the patients according to the median dialysis vintage in two groups: <66.5 and ≥66.5 months.
Supplementary table 1a: Fractional IL-10 translation (Ct values, basal) in HD patients stratified for the median dialysis time <66.5 and ≥66.5 months
|
|
F5 |
F6 |
F7 |
F8 |
F9 |
bG |
|
HD (<66.5) |
6.4±1.5 |
5.6±1.2 |
5.5±1.2 |
5.4±1.6 |
5.1±1.0 |
6.4±1.2 |
|
HD (≥66.5) |
6.6±3.5 |
5.4±2.1 |
5.2±1.7 |
5.4±1.4 |
5.4±1.6 |
6.9±1.4 |
|
ANOVA (adjusted p value) |
0.999 |
0.999 |
0.999 |
0.999 |
0.999 |
0.999 |
Supplementary table 1b: Fractional Casp-8 translation (Ct values, basal) in HD patients stratified for the median dialysis time <66.5 and ≥66.5 months
|
|
F5 |
F6 |
F7 |
F8 |
F9 |
bG |
|
HD (<66.5) |
0.3±1.5 |
-0.74±1.0 |
-0.3±0.9 |
0.5±0.9 |
0.9±0.8 |
2.4±0.8 |
|
HD (≥66.5) |
-0.2±2.1 |
-0.8±1.6 |
-0.3±1.2 |
0.5±0.8 |
1.0±0.7 |
2.7±0.4 |
|
ANOVA (adjusted p value) |
0.999 |
0.999 |
0.999 |
0.999 |
0.999 |
0.999 |
We added the result section:
3.6. Influence of dialysis-related parameters with regard to monosomal/polysomal IL-10 and Casp-8 expression
Dialysis-related parameters like dialysis vintage, Kt/V, or the kind of dialysis access (arteriovenous fistula versus catheter) may have an impact on the monosomal/polysomal IL-10/Casp-8 expression. Analyzing the data this way we did not see any effect. Neither dialysis vintage, Kt/V not the kind of dialysis access were related to IL-10 or Casp-8 expression (see supplementary data).
Lines 288 - 294.
Point 2: The Reviewer asks if the fractional translation data can be stratified according to kt/V and dialysis access (AvF/catheter) or method of dialysis (online, HS
Response 2: The Reviewer is right. Kt/V (a), dialysis access (b) and method of dialysis (c) may be associated with effects on anti-inflammation parameters.
- Kt/V analysis:
Supplementary table 2a: Fractional IL-10 translation (Ct values, basal) in HD patients stratified for the median kt/V value (<1.7 and ≥1.7)
|
|
F5 |
F6 |
F7 |
F8 |
F9 |
bG |
|
HD (<1.7) |
6.7±3.1 |
5.6±1.4 |
5.2±1.4 |
5.4±1.7 |
5.1 |
6.5±1.5 |
|
HD (≥1.7) |
6.3±2.2 |
5.4±1.9 |
5.4±1.5 |
5.6±1.4 |
5.4±1.4 |
6.8±1.2 |
|
ANOVA (adjusted p value) |
0.999 |
0.999 |
0.999 |
0.999 |
0.999 |
0.999 |
Supplementary table 2b: Fractional Casp-8 translation (Ct values, basal) in HD patients stratified for the median kt/V value (<1.7 and ≥1.7)
|
|
F5 |
F6 |
F7 |
F8 |
F9 |
bG |
|
HD (<1.7) |
0.08±1.7 |
-0.6±1.2 |
-0.3±0.9 |
0.3±1.0 |
0.9±0.8 |
2.5±0.8 |
|
HD (≥1.7) |
0.1±2.0 |
-0.9±1.5 |
-0.4±1.1 |
0.6±0.80 |
1.1±0.6 |
2.6±0.5 |
|
ANOVA (adjusted p value) |
0.999 |
0.999 |
0.999 |
0.999 |
0.999 |
0.999 |
We added the following sentences to the result section:
See Result section 3.6 and supplementary data.
- Dialysis access analysis:
HD patients were stratified according to permanent catheter (Pcat) or fistula (F) access
Supplementary table 3a y: Fractional IL-10 translation (Ct values, basal) in HD patients stratified according to permanent catheter (Pcat) or fistula (F) access
|
|
F5 |
F6 |
F7 |
F8 |
F9 |
bG |
|
Pcat |
7.5±3.9 |
5.6±1.2 |
5.5±1.1 |
5.0±1.9 |
5.2±1.2 |
6.5±1.6 |
|
F |
6.2±2.1 |
5.5±1.8 |
5.3±1.6 |
5.6±1.3 |
5.3±1.4 |
6.7±1.2 |
|
ANOVA (adjusted p value) |
0.999 |
0.999 |
0.999 |
0.999 |
0.999 |
0.999 |
Supplementary table 3b: Fractional Casp-8 translation (Ct values, basal) in HD patients stratified according to permanent catheter (Pcat) or fistula (F) access
|
|
F5 |
F6 |
F7 |
F8 |
F9 |
bG |
|
Pcat |
0.2±2.0 |
-0.8±1.7 |
-0.2±1.1 |
0.08±1.2 |
0.9±1.0 |
2.3±0.8 |
|
F |
0.07±1.8 |
-0.8±1.3 |
-0.4±1.0 |
0.6±0.7 |
1.0±0.6 |
2.7±0.6 |
|
ANOVA (adjusted p value) |
0.999 |
0.999 |
0.999 |
0.999 |
0.999 |
0.999 |
We added the following sentences to the result section:
See Result section 3.6 and supplementary data.
- Method of dialysis (online versus HD)
HD patient have been treated with polysulfone (N=41) or celluloses dialyzers (N=1), no patient was treated with online hemofiltration.
Point 3: The Reviewer asks if we can also compare samples from peritoneal dialysis patients with the method used.
Response 3: The Reviewer is right it would be interesting to have fractional Il-10 and Casp-8 expression data from peritoneal dialysis patients. This study did not address in how far different dialysis treatment modalities influence the fractional IL-10 expression analysis. We think, we have to investigate PD patients in a new study.
Point 4: The Reviewer indicates that image quality on page 5 should be improved (polysome separation curves)
Response 4: We are sorry for the low image quality of the polysomal separation curves. We try to insert a picture with a better resolution quality.
